# The Childhood Cancer Centre Is Coming Home: Experiences of Hospital-Based Home Care

**DOI:** 10.3390/ijerph18126241

**Published:** 2021-06-09

**Authors:** Malin de Flon, Gisela Glaffey, Linda Jarl, Kristin Sellbrant, Stefan Nilsson

**Affiliations:** 1Institute of Health and Care Sciences, Sahlgrenska Academy, University of Gothenburg, Box 457, 405 30 Gothenburg, Sweden; malin.de.flon@gmail.com (M.d.F.); gisela.glaffey@dinvcilerum.se (G.G.); linda.jarl@vgregion.se (L.J.); zence_88@hotmail.com (K.S.); 2University of Gothenburg Centre for Person-Centred Care, Sahlgrenska Academy, University of Gothenburg, Box 457, 405 30 Gothenburg, Sweden

**Keywords:** cancer, children, experience, home care, parents

## Abstract

Today, there is a shift towards care being given closer to the patient, with more children receiving care in their homes. Care at home has proven to be a viable alternative to hospital care, as shown by a project for hospital-based home care conducted in West Sweden. The aim of this study was to describe how children with cancer and parents experienced receiving care at home. After purposive sampling, six children with cancer aged 6–16 and eight parents participated. Semistructured interviews were performed, and the data were analysed using qualitative content analysis. Four main categories emerged: save time and energy in the family; maintain everyday life; feel trust in the healthcare professionals; mixed feelings about getting treatment at home. This hospital-based home care project created good conditions for both children with cancer and their parents to feel secure. In addition, home care can be very child-centric, whereby the caregivers involve the children by taking their thoughts and utterances into account.

## 1. Introduction

In Sweden, approximately 370 children suffer from cancer each year [1]. The diagnosis of cancer can be experienced as chaotic for the child and the family [2]. For most children and families, a cancer diagnosis leads to a transition to unpredictable everyday life with many hours in hospital [2,3]. However, hospital-based home care can be offered as a viable alternative to hospital care. In previous research, parents experienced this as a calmer and more predictable solution, and more compatible with family life, than being forced to stay in hospital [4]. In addition, parents can often be financially affected by hospital stays [5,6] and home care can strengthen family life without negatively impacting on the family’s finances due to, for example, travel costs [5]. Children have also reported advantages with conducting cancer treatment at home [7]. Consequently, hospital-based home care contributes to symptom control for children and increases quality of life for families [8].

Nilsson et al. [9] found that a common reason for home care is palliative care, which is probably more suitable to give at home than acute care. Successful home care requires skilled healthcare professionals who are able to relieve the child’s symptoms. Other research has also stressed the importance of managing the child’s symptoms [10]. However, healthcare professionals tend to miss highlighting psychological issues, and more often prioritize physical symptoms, e.g., pain and/or fatigue [11].

Today there is a shift towards care being given closer to the patient. In Sweden, receiving home care has become increasingly common and is expected to become even more frequent in the future [12]. Nurses have reported home care for children as challenging on both a professional and a personal level [13]. Nurses in community-based home care felt insecure about having to conduct care for children at home and asked for guidance on how to do this safely and successfully [14]. An alternative to community-based home care could be hospital-based home care, with nurses skilled in paediatric care [15]. Spiers et al. [16] showed that home care for children was both possible and as effective as hospital care. According to Hansson et al. [17], hospital-based home care is a possible alternative to hospital care for children with cancer, as it can be performed safely and with high patient satisfaction.

Care at home has proven to be an alternative to hospital care, as shown by a project for hospital-based home care conducted in West Sweden. For this project, there was a need to describe how children with cancer and their parents wanted their home care delivered, i.e., what things were experienced as important to them. This study is a first step to describe what topics children and parents find most important to explore. There are few studies from the child’s perspective, which makes each child’s voice important. In addition, home care is unusual, and children are often so ill or young that large groups are difficult to gather for this type of study. The purpose of this study was, therefore, to use a qualitative design to illuminate how children with cancer and their parents experience home care.

## 2. Materials and Methods

### 2.1. Study Design

A qualitative study design was conducted that described how children with cancer and their parents experience receiving home care. We used the COREQ 32 item checklist to report the research.

### 2.2. Setting

This study was conducted as part of a project on hospital-based home care for children with cancer. The project, which was a pilot study, was based on an evaluation of the needs in healthcare. The pilot study used the Plan-Do-Study-Act model to perform and evaluate home care for families where the child has cancer. The project was evaluated with questionnaires and qualitative interviews (in this article we present only the qualitative data). Physicians and nurses specialized in paediatric care, whose ordinary job was conducted at a childhood cancer centre in West Sweden, went to the homes of children with cancer. The main purpose of the project was to evaluate whether home care could be implemented with maintained patient safety and increased patient satisfaction. The COVID-19 pandemic affected people in Sweden while the project was being conducted, and the shift from hospital care to home care became even more important in order to avoid the spread of COVID-19.

### 2.3. Sample

Children with cancer and their parents were asked by the healthcare professionals if they wanted to participate in the study. Inclusion criteria were (1) children (aged 6–16 years) with cancer and their parents, and (2) that the participants spoke and understood Swedish or English. The children and their parents participated in the same hospital-based home care project, with four of the children and five of the parents coming from the same family.

### 2.4. Data Collection

An inductive qualitative approach was used to describe each participant’s feelings and experiences. The purpose of the topics was to qualitatively describe how the participants in this pilot study experienced receiving home care. The interviewers (M.d.F, G.G., L.J. and K.S.) were specializing to become district nurses and conducted the interviews. The interviews were conducted digitally via Zoom [18], which was an appropriate way to collect qualitative data due to the social distancing requirements of the prevailing COVID-19 pandemic. Interview length was determined by the participants. The children’s interviews took 20–50 min (mean 29 min), and the parents’ interviews took 17–31 min (mean 19 min). The interviews were conducted between June–October 2020.

In this study, photo voice was used for the children. This is a qualitative method developed in the 1990s by Wang and Burris [19], the idea being that images and words can effectively express a person’s needs, problems and desires. Carter and Ford [20] believe that opportunities are created to gain an increased understanding of the child’s own perception of the world and what feelings and thoughts the child has about his/her experiences of healthcare. Before the interviews, the children were asked to take photographs of things in the home that meant something to them. The interviews then started with the children being asked if they had taken any pictures they wanted to show and talk about. One child chose not to take any photographs at all, while the other children took one to nine photos each. A semistructured interview guide was then used.

The parents were asked semistructured questions about their experiences of receiving hospital-based home care. Each question started with “Tell us your thoughts about…”. The content of the interview guide was the home, the hospital, the healthcare professionals, participation and activities in everyday life.

The basic idea was that the interviews would take place with only the child in the room and a parent close to hand outside. However, the child’s wishes and need for security were allowed to govern, which meant that for all the interviews except one, one or both parents were present. However, the focus was still maintained on the children and their own experiences. The parents were interviewed individually, without their children.

### 2.5. Pilot Study

The interviewers (M.d.F, G.G., L.J. and K.S.) each conducted a pilot interview to evaluate whether the method of data collection was appropriate for the purpose of the study. G.G. and L.J interviewed the children, and M.d.F and K.S. interviewed the parents. The purpose of the pilot interviews was to test the interview guide, technical equipment and Zoom. The semistructured questions and relevant open-ended follow-up questions were considered to lead to data relevant to the purpose of the study, and no changes were made to the interview guides. The pilot interviews were, therefore, included in the results of the study.

### 2.6. Data Analysis

The interviews were recorded and transcribed verbatim by the interviewer who conducted each interview. The transcribed interviews were read by the interviewers several times to get an overview of the content. Data were interpreted through inductive content analysis, which is described as a way of working with the text where different levels of interpretation are made [21,22,23]. The process of analysis involves moving back and forth between parts of the text and the text as a whole, the focus throughout being on the content, i.e., what emerged visibly and clearly in the text [22].

The interviewers (M.d.F, G.G., L.J. and K.S.) selected meaning units, which were sentences, words or paragraphs that contained aspects related to the purpose of the study. The meaning units were subsequently condensed, which is a process in which the core of the text is retained while being shortened. Codes were created and compared for similarities and differences before being interpreted by creating categories. The last author (S.N.) read and reviewed the codes and discussed the categories with the other authors.

In the last step of the analysis, a triangulation was performed between the children’s experiences and the parents’ experiences. This triangulation eventually gave rise to themes that covered both the children’s and parents’ experiences. There were several common aspects, even if some experiences differed between the groups. We first present the children’s findings, and after these the parents’ findings, making it possible for the reader to find the main outcome for each group.

### 2.7. Ethical Considerations

The study was carried out in accordance with the ethical guidelines laid down in the Helsinki Declaration [24]. The participants received oral and written information about the study, and this was followed by parental consent and child assent. All participants were informed that their participation was voluntary and that they could withdraw from the study at any time without consequences and without having to state a reason. All information was kept confidential, and it is not possible to identify any individual person. An ethical review of the study has been conducted at the university that was responsible for the study.

## 3. Findings

Thirteen children and 16 parents were invited to participate in the study. A total of six children (five boys and one girl) aged 6–16 years participated. The children in the interviews had brain tumours, leukaemia and skeletal cancer. Eight parents from six different families (five mothers and three fathers) aged 28–50 years also participated in the study. Seven of the parents had a university level education and all were employed. These parents’ children had brain tumours, leukaemia and osteogenic sarcoma.

The results showed that the children and parents often had common thoughts about home care. Saving time and energy to maintain everyday life were common thoughts, and also the need for trust in the healthcare professionals. However, there were mixed feelings about having the healthcare professionals at home. The children liked this, but the parents sometimes liked going to the hospital more.

### 3.1. Save Time and Energy in the Family

The children thought that home visits took less time than hospital visits. The session at the hospital involved a long wait, for example, while having blood samples taken and then waiting for a physician or a nurse.

“… you have to stay a long time and you have to wait until… and then you have to wait when you book a taxi again so it’s hard. When they come home, it’s easier, that’s kind of it. You save time and you don’t have to go to the hospital very much.” (Child, 15 years)

Traveling to and from the hospital was perceived as taking a lot of energy, and the children thought that home care was energy-saving. In addition, fatigue could be easier managed at home, by just lying in bed.

“… there’s more energy, that I don’t have to go all the way home or to the car and so, it’s much nicer to be home… then it’s nice to be home right away. And not having to go all the way home.” (Child, 12 years)

The parents explained how important it was for them to limit the number of times they had to go to the hospital, as they felt the trip to the hospital was often difficult for the child.

“Then not traveling so much was a great time saver… Even if they try to make the wait as short as possible at the hospital, everything takes a lot of time, and you have to wait for a nurse and a doctor…” (Parent 5, mother 42 years)

The parents also stated that they were able to work at home in a more satisfactory way when the child received care at home.

### 3.2. Maintain Everyday Life

The children´s everyday life could go on as usual, if they received home care. Home visits did not interrupt their ongoing activities, and they could continue to do things they wanted, such as playing video games.

“… it’s hard, but then when they come here, they flush (subcutaneous venous port) then they go home, and I do not miss lunch.” (Child, 15 years)

The parents described the opportunity to be at home as a luxury, especially since they had previously spent a lot of time in hospital.

“… having a child with cancer you become institutionalized very quickly, the hospital becomes like a second home… very good because you need to be there so much… But the aim is always that life at home and everyday life should be so much bigger than the time at the hospital… then it’s a positive thing that they get to meet us in our real environment” (Parent 1, mother 28 years)

### 3.3. Feel Trust in the Healthcare Professionals

The children felt secure about receiving home visits if they knew the healthcare professionals. A good relationship with them made the child feel trust in the care situation.

“…when there’s someone you don’t know, you know nothing about them. So, you don’t know if they are too strict or something like that.” (Child, 9 years)

It was important to the children that the nurses were clinically skilled. For the children, this meant avoiding pain during procedures, and providing information and preparing them for what would happen. The children also wanted the nurses to be funny, helpful and kind, but still calm and determined, to show that they knew what to do.

“They’re nice. And strict in a good way… nurses should be tough… if they are not strict it takes a very long time because I’m a bit like this… I’m not ready for a while. So, then I need a tough guy who says you’re ready.” (Child, 9 years)

The parents wanted information about their child’s condition and treatment and felt that receiving home care gave an opportunity for this, as the healthcare professionals had time to answer questions because they only had to care for one patient at a time. The parents perceived the healthcare professionals to be less stressed, and consequently, that they took their time in a different way to when they were in hospital.

“It’s always felt like they have time… you’re able to chat a little and ask some questions … just positive and just as professional to meet them at home as in hospital, where we’ve absolutely not experienced any difference, only positives in this case.” (Parent 2, mother 46 years)

When the parents knew the healthcare professionals, it was easier to relax and feel safe, both in the parent-healthcare professional relationship and the child-healthcare professional relationship.

“… that there’s continuity with who comes… that there shouldn’t be too many different nurses who come in these teams, but yes, we have a few, and we know them… they are the closest to the child, but I think it’s very important that they try to keep continuity.” (Parent 8, mother 46 years)

### 3.4. Mixed Feelings about Getting Treatment at Home

The children described closeness to family and friends as being valuable when receiving home care. The child´s own room and bed were places that he/she believed created security. Pets could be involved in procedures at home, and stuffed animals could have a comforting and reassuring effect.

“… usually before I start my home treatment, I go out and tell them and they wish me luck and then I talk to them a bit, then it goes well. So, they kind of help me… They (the pets) have to wait for me to get ready and *Names of stuffed animals* have to lie down and comfort me.” (Child, 9 years)

The procedures were performed in different places in the home, for example, in the kitchen or on the sofa in the living room, depending on what the children wanted. One child said that it was nice to prepare a place in the home before the visit, for example, by arranging a lying position with pillows that made it easier. The children described their experiences of receiving care at home as a feeling of joy and satisfaction.

“… I’d rather be home… it was nicer to be home than to go there (to the hospital).” (Child, 12 years)

The parents had mixed reactions and experiences of having healthcare professionals at home. Some parents felt that home is home and in the hospital the child is sick. They felt that their home was their castle, a place where they could be free from all the sickness.

“… this is our home; this is our castle. We go to you and do these horrible things, and then we go home…” (Parent 6, mother 45 years)

One parent felt stressed by having healthcare professionals at home, as the house needed to be clean and in order. The parent felt it would have been easier to go to the hospital and avoid having the healthcare professionals at home.

The parents also mentioned they missed positive things about being in hospital, e.g., meeting other families, play therapy, clowns and other healthcare professionals. They experienced that they received less support with care in the home than with hospital care.

“… it’s the whole thing of not getting… the clowns, the other nurses… other parents… plus that I still have to wait for them (the team). They may be a little late or come a little too early… now you’re still sitting and waiting that half hour, I could just as easily have been at the hospital…” (Parent 8, mother 46 years)

The parents felt that it was easier to receive support from a social worker or a psychologist when they were offered this at the hospital. They experienced receiving care in hospital as positive because the healthcare professionals were easily accessible. They also felt positive about always having access to a physician at the hospital.

## 4. Discussion

There were similarities in the experiences of hospital-based home care for both the children and parents in our study. Both groups felt they saved time and energy by staying at home. However, the parents sometimes missed proximity to other healthcare professionals, such as social workers or psychologists. The parents also had more mixed feelings about home visits than the children.

In our study, the children felt positive about having their everyday routines maintained through home care. Finding strategies to save energy for children with cancer is important, as illustrated by one study where more than half of the children with cancer reported challenges with fatigue at home [25]. Fatigue affects children’s opportunities to participate in everyday activities [26]. Strategies to maintain energy include resting, engaging in quiet activities and changing schedules to be more appropriate for the situation [27].

The parents in our study felt that home care gave them a chance for a normal family life after spending a lot of time in the hospital. In hospital, only one parent was allowed due to the COVID-19 pandemic. In another study, parents described home care as providing a sense of freedom during a stressful period in the family´s life [13]. This is similar to our findings, where the parents felt receiving treatment for their child at home was an opportunity to help the day go smoother, as the family could continue their activities and spend more time together.

The results of our study and previous research emphasise that healthcare professionals should be interested and able to adapt care actions to the child’s daily form [28]. This is further reinforced by Farjou et al. [29], who state children consider the attitude of the healthcare professionals and how they communicate to be important. It turned out that many children wanted the healthcare professionals to be friendly, helpful and caring and have a positive attitude.

Trust was something that both children and parents highlighted as important in hospital-based home care. The children stressed that the healthcare professionals had to be skilled, and the parents had to be convinced that the care had been fulfilled satisfactorily. The need for trust has also been highlighted in previous research, where the parents felt it was important to trust and have confidence in the healthcare professionals’ care [4]. Trust in healthcare professionals is not self-evident and needs to be built up over time [30]. However, parents often choose to trust healthcare professionals to make the right decisions [31].

The parents stated that with home care, it became tricky to plan their day’s activities. Sometimes they had to wait for the healthcare professionals from the hospital and felt that it would have been easier to go to the hospital instead. This is in line with the findings of Molinaro and Fletcher [32] who revealed that parents need to reprioritize their time in order to adapt to the new situation. They found it difficult to juggle the demands of life. Thus, parents of children with cancer need support to adapt to the new situation [3].

The children in our study thought it was important to be with their parents and that closeness to other relatives and friends also created joy and peace. This agrees with previous findings that highlight the family as important for the child with cancer [33]. Another study also stressed that most children with cancer experienced a varying degree of social exclusion. Whenever possible, it is important to maintain social relationships for children with cancer [34], as social distance from friends contributes to psychosocial illness [35]. Healthcare professionals’ care should thus support the child’s functioning in everyday life [36].

Our study highlighted that pets gave the children closeness and friendship. If the children were sad, the pets could help them feel happy again. The importance of a pet has also been confirmed in previous research [37,38] and stresses the therapeutic effects of having animals in care situations [39].

Our findings suggest that there are many benefits of offering hospital-based home care to children with cancer and their parents and reinforces the findings of Castor et al. [13] that stress the values of offering children and their families care in their homes. Castor et al. [40] also confirmed that the acceptability of home care was high for children in different stages of illness, including end of life.

Our study has some limitations that influence the findings. The small sample size of participants and the fact that the study was conducted in only one childhood cancer centre at a university hospital could limit the credibility of the study. Due to this limitation, the findings might not be representative of all groups of children with cancer and their parents. Another limitation was that, of the 13 children, four came from the same family, and of the 16 parents, five came from the same family. The incorporation of parents and children from the same families may have affected the findings. However, due to the small sample size, we judged that this was the best solution. The study was strengthened by more participants who shared their experiences. Despite this, the limited number of participants emphasizes the need to follow up our preliminary findings with more studies which include a larger sample size.

## 5. Conclusions

The findings of this study show that by receiving care at home, the children with cancer and their parents saved time and energy and could maintain family life. The children appreciated being close to their own things and staying in a safe home environment with their family. The children and the parents wanted to meet healthcare professionals they already knew because it was perceived as creating a sense of security. In addition, the parents wanted to feel trust in the healthcare professionals. It was desirable for the children that the healthcare professionals took the child’s perspective, for example, by involving them in their care.

## Data Availability

The data used and/or analysed during the current study are available from the corresponding author on reasonable request.

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
