# Peer review of "The Childhood Cancer Centre Is Coming Home: Experiences of Hospital-Based Home Care"

_ijerph, 2021, doi:10.3390/ijerph18126241_

Round 1

Reviewer 1 Report

Pediatric home care (PHC) facilitates continuation of a normal life for children and their families. This paper presents a qualitative assessment about how children with cancer and their parents wanted their home care delivered. Thus, the aim of this study is useful to improve children with cancer quality of life and it is original since it considers patients’ psychological aspects.

Some comments:

Materials and methods.

  • Authors should follow Consolidated criteria for reporting qualitative research (COREQ): a 32-item checklist for interviews and focus groups, to report accurately the research.
  • There are some surprising terms in a scientific article. For instance, in the study design section, ‘A qualitative study design was conducted that illuminated how…’ (line 61).
  • Setting: Since this study was conducted as part of a project on hospital-based home care, authors should describe this project with more detail.
  • Sample: This part should be at the beginning of result section.

In addition, out of the 13 children, four came from the same family, and of the 16 parents, 5 came from the same family. How did it influence in the results? Why did they decide include children from the same family? An explanation should be in the limitation section.

  • Data collection: researchers carried out semi-structured questions. To clarify this point, authors could include more detail in the different topics include. In addition, they could explain why they decided these topics: did the authors carry out a previous systematic review about the main topics to be asked?

Researchers:

  • Authors have described the main findings through the transcription of children and parents’ opinions. However, I miss some information about cross-validation of the findings (even if there are few patients included).

Author Response

Reviewer 1

Reviewer´s comment

Authors´ reply

Materials and methods.

Authors should follow Consolidated criteria for reporting qualitative research (COREQ): a 32-item checklist for interviews and focus groups, to report accurately the research.

We have used the COREQ 32 item checklist to report the research.

There are some surprising terms in a scientific article. For instance, in the study design section, ‘A qualitative study design was conducted that illuminated how…’ (line 61).

We have reworded this sentence, to: “A qualitative study design was conducted that described how children with cancer and their parents experience receiving home care.”

Setting.

Since this study was conducted as part of a project on hospital-based home care, authors should describe this project with more detail.

We have added text about the main project, i.e., “The project, which was a pilot study, was based on an evaluation of the needs in healthcare. The pilot study used the Plan-Do-Study-Act model to perform and evaluate home care for families where the child has cancer. The project was evaluated with questionnaires and qualitative interviews (in this article we present only the qualitative data).”

Sample.

This part should be at the beginning of result section.

We have moved the text to the beginning of the findings.

In addition, out of the 13 children, four came from the same family, and of the 16 parents, 5 came from the same family. How did it influence in the results? Why did they decide include children from the same family? An explanation should be in the limitation section.

We have written in the limitation “Another limitation was that, of the 13 children, four came from the same family, and of the 16 parents, 5 came from the same family. The incorporation of parents and children from the same families may have affected the findings. However, due to the small sample size, we judged that this was the best solution. The study was strengthened by more participants who shared their experiences..”

Data collection.

Researchers carried out semi-structured questions. To clarify this point, authors could include more detail in the different topics include. In addition, they could explain why they decided these topics: did the authors carry out a previous systematic review about the main topics to be asked?

The purpose of the topics was to qualitatively describe how the participants in this pilot study experienced receiving home care.

Researchers

Authors have described the main findings through the transcription of children and parents’ opinions. However, I miss some information about cross-validation of the findings (even if there are few patients included).

In the last step of the analysis, a triangulation was performed between the children's experiences and the parents' experiences. This triangulation eventually gave rise to themes that covered both children's and parents' experiences. There were several common aspects, even if some experiences differed between the groups.

Reviewer 2 Report

de Flon et al. describe an interview-based survey of patients and their parents regarding home-based care for pediatric cancer. These programs are interesting and are undoubtedly meaningful for patients with chronic illness. A process of informed consent is described however I do not see mention of a process of review by an insitutional review board. The manuscript is overall well written for grammar and fluidity, and the sections appear largely appropriately constructed for content with exception of the findings section, which contains no quantifiable summary data.

These programs are interesting however the construction of this investigation makes it difficult to assess effects of the intervention. The shared sentiments from participants are heavily edited for content and largely not objective.

It is difficult to understand what the study population was. The description in section 2.3 Sample could be better organized, as mentioning who was invited first instead of in the middle of the section would be better. It is difficult to consider parent and child responses as equivalent in the same sample and these populations should have been separated in the analysis. It is not clear whether all participants were represented equally in the results. The inclusion of 4 children from the same family also raises questions regarding whether the sample was sufficiently large and varied.

Given these issues I would recommend that the authors approach this question from a different perspective examining only quantifiable metrics, and comparisons to a control would be important. A control could be a historical/current population, the prior experience of the participants, etc. A larger sample size is necessary as well. Without a more objective approach to this question the impact of this intervention is unknown.

Author Response

Reviewer 2

de Flon et al. describe an interview-based survey of patients and their parents regarding home-based care for pediatric cancer. These programs are interesting and are undoubtedly meaningful for patients with chronic illness. A process of informed consent is described however I do not see mention of a process of review by an institutional review board.

An ethical review of the study has been conducted at the university that was responsi-ble for the study.

These programs are interesting however the construction of this investigation makes it difficult to assess effects of the intervention. The shared sentiments from participants are heavily edited for content and largely not objective.

The purpose of the study was to interpret the participants’ experiences, due to that it is not possible to assess effects in a qualitative interview. However, even if all qualitative analyses in some way are subjective, it is our intention to reach trustworthiness by strictly following the rules for qualitative analysis. I.e., our ambition is to reach credibility and transferability.

It is difficult to understand what the study population was. The description in section 2.3 Sample could be better organized, as mentioning who was invited first instead of in the middle of the section would be better. It is difficult to consider parent and child responses as equivalent in the same sample and these populations should have been separated in the analysis. It is not clear whether all participants were represented equally in the results. The inclusion of 4 children from the same family also raises questions regarding whether the sample was sufficiently large and varied.

The children and the parents were analyzed separately in the first step. In the next step, we did a triangulation where we see that several aspects were in common. We chose to present equalities and differences in the findings, and to present the findings together. However, we first present the children´s findings, and after this the parents´ findings, i.e., it is possible for the reader to find the main outcome for each group.

Given these issues I would recommend that the authors approach this question from a different perspective examining only quantifiable metrics, and comparisons to a control would be important. A control could be a historical/current population, the prior experience of the participants, etc. A larger sample size is necessary as well. Without a more objective approach to this question the impact of this intervention is unknown.

Thank you for this suggestion. However, the intention with the data material was to describe the participants´ experiences and the analysis is inductive. We have compared the experiences from the participants with previous qualitative studies in the discussion. The references in the discussion confirm that our themes can be transferable.

Reviewer 3 Report

This is a good thematic. This context of care is growing.

The text on page 2, line 79 and 82, it is repeated, consider reviewing. 

Why didn't you use software to support qualitative data analysis?

Ethical considerations - Consider including approval by the ethics committee.

Consider presenting introductory text to the findings.

Author Response

Reviewer 3

The text on page 2, line 79 and 82, it is repeated, consider reviewing.

We have tried to reword the sentence “Thirteen children and 16 parents were invited to participate in the study. A total of six children (five boys and one girl) aged 6-16 years participated. The children in the interviews had brain tumours, leukaemia, and skeletal cancer. Eight parents from six different families (five mothers and three fathers) aged 28-50 years also participated in the study. Seven of the parents had university level education, and all were employed. These parents’ children had brain tumours, leukaemia and osteogenic sarcoma.”

Why didn't you use software to support qualitative data analysis?

The choice of software (such as Atlas.ti and NVivo) can be valuable when you have large data samples. However, in this case we found the data suitable to handle with transcribed text in a word document. The text was read back and forth and meaningful codes were chosen from the text. These codes were then used in the further analysis.

Consider presenting introductory text to the findings.

We have added text to the introduction of the findings, i.e., “The results showed that the children and parents often had common thoughts about home care. Saving time and energy to maintain everyday life were common thoughts and also the need for trust in the healthcare professionals. But there were mixed feelings about having the healthcare professionals at home. The children liked this but the parents sometimes liked going to the hospital more.”

Round 2

Reviewer 1 Report

Authors have included all the comments in the new version.

Author Response

Reviewer 1

Reviewer´s comment

Authors´ reply

Authors have included all the comments in the new version.

Thank you!

Reviewer 2 Report

I appreciate the authors' attention to the reviewers' comments. Given the small sample size and lack of quantitation or comparison of any indices I am not sure what to make of this paper aside from its sharing opinions of a few program participants, almost a third of which were in the same family and therefore likely skewed the responses. Without any scientific comparisons it is impossible to say whether this intervention is an overall improvement or detriment compared to current care.

Author Response

Reviewer 2

I appreciate the authors' attention to the reviewers' comments. Given the small sample size and lack of quantitation or comparison of any indices I am not sure what to make of this paper aside from its sharing opinions of a few program participants, almost a third of which were in the same family and therefore likely skewed the responses. Without any scientific comparisons it is impossible to say whether this intervention is an overall improvement or detriment compared to current care

Thank you for your comment.

We have added in the introduction “This study is a first step to describe what topics children and parents find most important to explore. There are few studies from the child's perspective, which makes each child's voice important. In addition, home care is unusual and children are often so ill or young that large groups are difficult to gather for this type of study. The purpose of this study was therefore to use a qualitative design to illuminate how children with cancer and their parents experience home care”, and in the limitations “Despite this, the limited number of participants emphasizes the need to follow up our preliminary findings with more studies which include a larger sample size.”.